# Machine Learning to Identify Interaction of Single-Nucleotide Polymorphisms as a Risk Factor for Chronic Drug-Induced Liver Injury

**DOI:** 10.3390/ijerph182010603

**Published:** 2021-10-10

**Authors:** Roland Moore, Kristin Ashby, Tsung-Jen Liao, Minjun Chen

**Affiliations:** Division of Bioinformatics and Biostatistics, National Center for Toxicological Research, U.S. Food and Drug Administration, 3900 NCTR Rd, Jefferson, AR 72079, USA; rolmoore2000@gmail.com (R.M.); kristin.mceuen@fda.hhs.gov (K.A.); Tsung-Jen.Liao@fda.hhs.gov (T.-J.L.)

**Keywords:** drug-induced liver injury, chronicity, machine learning, SNP, genotype, gene–gene interactions, epistasis, splines

## Abstract

Drug-induced liver injury (DILI) is a major cause of drug development failure and drug withdrawal from the market after approval. The identification of human risk factors associated with susceptibility to DILI is of paramount importance. Increasing evidence suggests that genetic variants may lead to inter-individual differences in drug response; however, individual single-nucleotide polymorphisms (SNPs) usually have limited power to predict human phenotypes such as DILI. In this study, we aim to identify appropriate statistical methods to investigate gene–gene and/or gene–environment interactions that impact DILI susceptibility. Three machine learning approaches, including Multivariate Adaptive Regression Splines (MARS), Multifactor Dimensionality Reduction (MDR), and logistic regression, were used. The simulation study suggested that all three methods were robust and could identify the known SNP–SNP interaction when up to 4% of genotypes were randomly permutated. When applied to a real-life DILI chronicity dataset, both MARS and MDR, but not logistic regression, identified combined genetic variants having better associations with DILI chronicity in comparison to the use of individual SNPs. Furthermore, a simple decision tree model using the SNPs identified by MARS and MDR was developed to predict DILI chronicity, with fair performance. Our study suggests that machine learning approaches may help identify gene–gene interactions as potential risk factors for better assessing complicated diseases such as DILI chronicity.

## 1. Introduction

As the primary organ responsible for metabolism, the liver is vulnerable to injury caused by drugs and drug metabolites [1,2]. Drug-induced liver injury (DILI) is one of the main reasons for halting drug development processes [3], and has caused over 50 approved drugs to be withdrawn from the market [4,5]. Moreover, even when a drug is deemed safe and approved for public use, a relatively small fraction of the population taking the drug may experience liver damage, and therefore identifying risk factors is of particular importance for preventing DILI [6,7]. The risk factors for DILI include age, sex, drug properties, and genetic variations [8,9].

Genetic factors have attracted increasing attention, and recent studies have shown that genetic variants in drug-metabolizing enzymes and human leukocyte antigen (HLA) were associated with the occurrence of DILI [10,11,12]. Many efforts have focused on the genetic risk factors of DILI, and several genetic variants such as single-nucleotide polymorphism (SNP) and insertion/deletion have been identified as risk factors associated with the use of specific drugs [13,14,15,16]. For example, carriers of HLA-B*57:01 are 80 times more likely to develop flucloxacillin-induced DILI than those who do not carry the variant [17]. The association of liver injury caused by specific drugs or groups of drugs with polymorphisms in HLA was also reported [18,19]. However, most of these studies investigated the association of each SNP individually [20,21]; SNP–SNP interaction as a potential risk factor for drug-induced liver injury was seldom reported.

Pairwise SNP–SNP interactions have been recognized to play a role in assessing disease susceptibility and drug responses [22,23]. Several statistical methods have been used to identify SNP–SNP interactions [24]. For example, Cook et al. [25] employed classification and regression trees and multivariate adaptive regression spline (MARS) models to explore the presence of genetic interactions for ischemic stroke. Moore and Williams [26] successfully applied the multifactor dimensionality reduction (MDR) method to identify gene–gene interactions in essential hypertension. Notably, SNPs themselves come in large numbers, sometimes in the hundreds of thousands; including their interactions, the number of factors which need to be considered could expand exponentially. The issue of dimensionality becomes a challenge of statistical genetic data analysis, especially when the sample size is relatively small compared to the number of factors. Certain SNPs tend to occur together and are not statistically independent, which can complicate analysis. The combination of SNPs that are preferably uncorrelated to each other may improve the classification performance [27]. Statistical genetic data analysis is generally computationally intensive and time-consuming, and classical statistical methods such as regression are less feasible for use with such high-dimensional data. Thus, appropriate statistical/machine learning methodologies are critical for overcoming the challenges of analyzing gene–gene interactions.

In this study, we aim to investigate SNP–SNP interaction as a potential risk factor for predicting DILI chronicity using machine learning approaches. We briefly introduce three machine learning methods, including MARS, MDR, and logistic regression. Then, we use simulated data with known pairwise SNP–SNP interactions to evaluate the accuracies and robustness of these methods. Next, we apply the three methods to identify SNP interactions associated with chronic DILI. Finally, we evaluate the predictive performance of the identified SNPs by using a simple decision tree model to assess whether the presence of these SNP–SNP interactions is associated with an increased risk of DILI chronicity.

## 2. Materials and Methods

### 2.1. DILI Chronicity Cohort

Cases in this study were collected by International Serious Adverse Event Consortium (iSAEC), a large international collaborative study, including partnerships with the United Kingdom, Sweden, Spain, Germany, France, Switzerland, the Netherlands, Australia, and Finland. The study protocols were approved by local ethics committees, and the informed consent was obtained from all subjects involved in the study. Inclusion criteria for DILI cases followed the clinical chemistry criteria defined [28]. Specifically, a case must have either alanine aminotransferase elevated ≥5× the upper level of normal, or elevated alkaline phosphatase ≥2× the upper level of normal, or elevated levels of alanine aminotransferase ≥3× the upper level of normal while bilirubin concentrations are also two-fold higher than the upper level of normal. Causality assessment was conducted using the Roussel Uclaf Causality Assessment Method scoring system and expert review, consisting of a panel of three hepatologists. Only cases with a causality scale of probable with a score of greater than or equal to six were included. Cases with preexisting liver disease were excluded. DILI chronicity was determined by the time of patient recovery, which was defined as the days from the medication stop date until the date that the patient’s serum liver biochemistries returned to normal. For this dataset, chronic DILI is defined as liver injury without recovery for six months or more, termed chronic; acute DILI is defined as liver injury with recovery within six months, termed acute.

A dataset with 271 patients (33 chronic versus 238 acute DILI) were used to assess the three machine learning methods for identifying SNP–SNP interaction as a risk factor for chronic DILI. DNA preparation is described here [29] and standard quality control procedures were followed [30]. Samples were genotyped using the HumanOmniExpressExome-8v1 or the Illumina HumanCoreExome-12 v1.0 BeadChip, from which 872 SNPs associated with the genes in bile acid pathways were retrieved for analysis. Bile acid pathways were downloaded from the Molecular Signature Database (MSigDB) [31] and included pathways involved in bile acid synthesis, recycling, and transport.

### 2.2. Machine Learning Approaches to Identify SNP–SNP Interactions

Assuming *Y* is the phenotype, such as chronic DILI (binary; chronic or acute); P(Y=1)=p, *A* is an SNP, *B* is another SNP, and *A* × *B* is an SNP–SNP interaction factor, the parametric model for the logistic regression is as follows:(1)$$\log\left(\frac{p}{1-p}\right)=\beta_{0}+\beta_{1}A+\beta_{2}B+\beta_{3}A\times B$$
by contrast, the non-parametric statistical method model is:(2)Y=f(A,B)

The statistical methodologies for identifying gene–gene interactions were reviewed thoroughly [22,23]. Here, three frequently used methods including MARS [32], MDR [33,34], and logistic regression [35] were selected for evaluation based on availability and use.

#### 2.2.1. MARS Approach

The MARS approach is an adaptive algorithm for identifying SNP–SNP interactions, which fits models using flexible regression modeling, including non-linear components and interactions [32]. It was reported to perform better than typical logistic regression methods when applied to the discovery of interactions among genes without strong marginal effects [25].

In MARS, the entire dataset was divided into multiple smaller regression subsets through automatic dataset-determined knots. Each of these subsets comes with a basis function (usually a linear spline function), and all significant basis functions are aggregated to obtain the overall regression model. MARS gives a regression-like output model with certain basis functions of the predictors. It builds models of the following equation form:(3)$$\hat{f}=\sum_{i=1}^{k}C_{i}B_{i}\left(x\right)$$
where each $C_{i}$ is the coefficient multiplying a basis function $B_{i}\left(x\right)$, k is the number of knots, and $\hat{f}$ is the model estimation.

The regression model of MARS is derived directly from the data, through a data-driven automatic set of basis functions with their corresponding coefficients. These basis functions are derived based on automatic data-driven hinge values in the data, also referred to as knots. As a specific linear relationship between the response and predictors in a data subset is being modeled, the knot automatically identifies the point of direction change from that relationship. This point of change becomes a starting point for a new subset of another relationship, and the process continues to the end with new knots and relationships. In this way, the model captures both linear and non-linear relationships, with corresponding interactions.

The MARS model is built using a forward-and-backward selection process in the following way: the forward selection first combines all significant basis regression functions, and their interactions, and then backward selection prevents overfitting by pruning the result from the forward selection, removing basis functions one at a time, and selecting the model with the lowest Generalized-Cross-Validation (*GCV*) score. This process will yield the optimized final model. The *GCV* equation is written as follows:(4)GCV=∑i=1N(yi−f^)2(1−C/N)2
where C=1+p∗k with k as independent number of base functions and p as the penalty for adding a base function. Here, $y_{i}$ (i = 1 to N) is the value of phenotype for the observation i, and N is number of observations, and $\hat{f}$ is from the Equation (3).

#### 2.2.2. MDR Approach

The MDR approach is a non-parametric dimension-reducing method that was created primarily to detect gene–gene and/or gene–environment interactions [33,34,36]. Compared with MARS and other statistical methods which usually handle interactions between two factors, MDR has the power to identify statistically significant high-order interactions among three or more factors [33]. Its central concept is to utilize a specially designed strategy to transform multilocus information to a one-dimensional model. In this strategy, a subset of n genetic factors are first selected, and then these n-genetic factors and their possible multifactor classes are represented by n-dimensional space, in which each multifactor cell will be labeled as high-risk or low-risk group. In this way, the case-control model for multilocus genotypes will be converted into classifications of highrisk and low-risk, which reduces the n-dimensional model to a one-dimensional model. In the next step, the prediction error estimated through cross-validation is used to evaluate the selected n-genetic factor. For each n-genetic factor combination, a single model that minimizes the average classification error in the cross-validation training sets is selected. This will result in a list of best models, one for each value of n-genetic factor. Among these classification models, the combination of the genetic factors and the model they built that minimizes the average prediction error across the prediction errors in the cross-validation testing sets will be selected as the final one.

#### 2.2.3. Logistic Regression Approach

Logistic regression is the mostly used method for identifying gene–gene interaction; however, it is not so successful in its handling of the datasets with a large number of SNPs to consider, and weak or no marginal effects of SNPs [37]. Here, Random Forest was used together with logistic regression approach [38]. This approach used multiple decision trees as a learning ensemble to give a single output. Each tree acts on random bootstrap subsets of the data, as well as random subsets of the SNP predictors to make predictions. Each predictor represents the node of a tree, and a route links a sequence of predictors from the roots to the leaves. The predictions from each of these trees are then aggregated to give the overall output prediction. Majority voting is employed in classification. Random Forest produces a table ranking of the SNP predictors in the order of importance of their contributions to the phenotype. Logistic regression is then applied to the first few variables to determine the interacting SNPs that are linked to the phenotype.

### 2.3. Data Analysis

Here, we firstly employed a simulated dataset with known SNP–SNP interactions to evaluate the performance and robustness of these machine learning methods. The simulated dataset was modified from the dataset retrieved from the MDR software package. The dataset contains 250 observations, including 125 positive and negative phenotype responses, respectively, and 25 SNPs, of which SNP4 and SNP9 are a pair of known SNP interaction. The frequencies of the minority genotypes of SNP4 and SNP9 are 21% (52/250) and 23% (57/250). To test the robustness of the methodologies, we permutated the genotypes of 23 SNPs, except SNP4 and SNP9. In the first test, two of the 250 observations were selected for permutation, while in the second test 10 observations were permutated. Each analysis includes 100 permutations.

We also applied these machine learning approaches to a real-life DILI chronicity dataset. Figure 1 briefly illustrated the pipeline for identifying SNP–SNP interactions as potential risk factors which are associated with chronic drug-induced liver injury. Firstly, based on the dataset of 271 patients and 872 SNPs, three machine learning methods including MARS, MDR and Random Forest plus logistic regression were utilized to identify SNP–SNP interactions linked to DILI chronicity. Next, only the SNP–SNP interactions have the better associations with DILI chronicity, which is measured by odds ratio, than the individual SNPs were considered. Finally, The SNPs from the selected interactions were pooled together as candidate predictors, and then a decision tree model using classification and regression trees (CART) algorithm is developed.

All analyses except those specifically mentioned were performed using R (version 3.6.1) [39] and the MDR package [40] for multifactor dimensionality reduction approach, the randomForest package [41] for Random Forest algorithm, the Rpart package [42] for decision tree model, and the Stats package for logistic regression and Fisher exact test. Multivariate adaptive regression spline approach was implemented using the MARS engine of Salford Predictive Modeler 8.0 from MinTab [43].

## 3. Results

### 3.1. Simulation Analysis

Three machine learning methods were evaluated using the simulated dataset with a known SNP–SNP interaction (SNP4 + SNP9). As shown in Table 1, all three methods successfully identified the known SNP interaction. We further conducted a permutation analysis to test the robustness of the three methods. In the first test, the genotypes of two observations randomly selected from 250 observations were permutated. As a result, all three methods successfully identified the known SNP interaction. Even when the number of permutated observations was increased from 2 to 10 of 250 observations, MDR slightly decreased in recovery rate and correctly identified the known SNP–SNP interactions among 97% of 100 permutations, while logistic regression and MARS still maintained a 100% recovery rate of the known SNP–SNP interaction. In other words, these machine learning approaches were robust when up to 4% (or 10 of 250) of the observations were permutated.

### 3.2. Chronic DILI Data Analysis

We first checked the association of DILI chronicity with individual SNPs using the Fisher exact test. Five individual SNPs from SNP–SNP pairs selected by the machine learning methods below were tested. As shown in Table 2, two SNPs (rs6487213 and rs5417) were significantly associated with DILI chronicity, with odds ratios of 3.28 (95% CI: 1.57–7.09, *p* = 0.002) and 3.01 (95% CI: 1.44–6.43, *p* = 0.004), respectively. The other three cited SNPs were not statistically significant.

We also applied the three machine learning methods to identify the SNP interaction terms that were linked to DILI chronicity. As shown in Table 2, MARS identified an interaction of a two-SNP combination (rs6487213 + rs3785157), while RF-LR detected another two-SNP interaction (rs5417 + rs3785157), and MDR found a three-SNP interaction (rs5417 + rs7658048 + rs12453290). Importantly, the SNP interactions identified by MARS and MDR have an improved association between the presence of genotypes and the chronicity of DILI. The odds ratio of the SNP interaction rs6487213 + rs3785157 identified by the MARS approach increased to 4.74 (95% CI: 2.14–10.39, *p* < 0.001), while the odds ratios of rs6487213 and rs3785157 alone are 3.28 (95% CI: 1.57–7.09, *p* = 0.002) and 1.49 (0.72–3.14, *p* = 0.282), respectively. Additionally, 30.4% of the combined genotypes of CC + CC by rs6487213 + rs3785157 were reported with DILI chronicity, as compared to 20.8% of the genotype CC of rs6487213 alone. The MDR approach identified a three-SNP interaction, and its odds ratio of 4.19 (95% CI: 1.36–11.74, *p* = 0.008) by rs5417 + rs7658048 + rs12453290 was improved in comparison to the odds ratio of 3.01 (95% CI: 1.44–6.43, *p* = 0.004) by rs5417 alone. However, the SNP–SNP pair (rs5417 + rs3785157) identified by logistic regression did not show improvement over individual SNPs (Table 2).

We further utilized the SNPs identified by MARS and MDR to develop a simple decision tree model for predicting DILI chronicity. As shown in Figure 2, three SNPs (rs6487213, rs5417 and rs12453290) were selected as the predictors for the decision tree model built on 271 patients. The leaf defined by rs6487213, rs5417, with the genotype CC and AA, yielded a frequency of 36.4% (12/33) of DILI chronicity as compared to the overall frequency of 12% (33/271) in the study population. Other leaves defined by rs6487213, rs5417 and rs12453290, together yielded a frequency of 50% (3/6) DILI chronicity. When evaluated by using a stratified 5-fold cross-validation, the decision tree model yields a sensitivity of 42.4%, a specificity of 90.3%, an accuracy of 86.3%, and a balanced accuracy of 66.4%, which is calculated from the average value of sensitivity and specificity [44].

## 4. Discussion

Individual SNPs reportedly have limited predictive power on human phenotypes, especially for diseases with complicated mechanisms, such as DILI. Here, we investigated three different machine learning approaches: MARS, MDR, and Random Forest plus logistic regression, for their capabilities in identifying SNP–SNP interactions associated with DILI chronicity.

Logistic regression is a widely used approach for detecting SNP–SNP interactions; however, it has limited power to handle the exponentially growing interaction terms or the need of the large sample sizes to detect effects with high dimensions [26]. The combined use of Random Forest and logistic regression improved efficiency in searching SNP interactions but still could miss the best SNP–SNP interaction term by falling into local maxima. MARS and MDR are designed to overcome these limitations to identify the maximum effects of interaction terms in a moderate sample size [37,45]. MDR can detect the high order of SNP interactions, while MARS is faster in revealing candidate SNPs in the chronicity DILI dataset. Note, our DILI chronicity dataset only contains 872 SNPs and is not a large dataset. Regardless, both MARS and MDR successfully identified the SNP–SNP interaction terms with a higher predictive power than individual SNPs, while Random Forest plus logistic regression failed in this test.

A simple predictive model based on the SNPs identified by MARS and MDR also demonstrated a fair predictive performance, suggesting that the presence of these SNP–SNP interactions as risk factors could be associated with an increased risk of chronicity of drug-induced liver injury. We also explored possible underlying biological mechanisms associated with the SNPs identified by the machine learning approaches. Elevated total bilirubin was reportedly associated with DILI chronicity [46,47] and was confirmed in our study. Here, SNP rs6487213, a variant of gene SLCO1B1, was identified as an important contributing factor for predicting chronic DILI. SLCO1B1 is a solute carrier organic anion transporter family member 1B1. It is responsible for bilirubin uptake and transporting xenobiotic compounds such as toxins and drugs from blood into the liver, to be eventually excreted from the body. The uptake capability is markedly decreased by the presence of SLCO1B1 allelic variants. In a study involving 9500 Caucasians, the allelic variation in SLCO1B1 was reported as a major genetic predictor of increased serum bilirubin levels [48]. Furthermore, the polymorphisms of SLCO1B1 were reportedly associated with hepatoxicity caused by drugs, including but not limited to methimazole [49], methotrexate [50], atorvastatin [51] and anti-tuberculosis drugs [52], including rifampin [53].

Increasing evidence suggests that membrane transporters impact immune cell function and shape immunological responses. The SNP rs5417 is a variant of the gene GLUT4 (also known as solute carrier family 2, facilitated glucose transporter member 4, SLC2A4), which is a critical transporter for glucose uptake and metabolism. GLUT1 upregulation plays a key role in T cell activation and in importing glucose to be metabolized through aerobic glycolysis. Similar to GLUT1, GLUT4 was upregulated in activated CD4+ T cells to varying degrees, and the variants in GLUT4 play compensatory and supporting roles in most T cells [54].

Furthermore, the SNP rs12453290 is a variant of the gene MCT4 (SLC16A3), which is a member of the monocarboxylate transporter (MCT) family controlling the lactate uptake. MCT4 plays a role in exporting additional lactate into the extracellular space. Lactate often builds up at sites of inflammation and in hypoxic conditions. Excess lactate importing into T cells will restrict proliferation and cytokine production of cytotoxic T cells. In T cells, T cell receptor activation upregulates MCT1 expression, while its blockage will result in impaired cytotoxic T cell function. MCT4 upregulation causes full macrophage activation, while its deletion will lead to an intracellular decrease in glycolysis and accumulation of lactate. By contrast, the downregulation of MCT4 will promote the cytotoxicity of natural killer cells and boost immune responses [55].

## 5. Conclusions

In this study, we investigated three machine learning methods for the study of gene–gene interactions as risk factors for chronic DILI, including MARS, MDR and Random Forest followed by logistic regression. All of the three methods were robust when tested in the simulation dataset with permutation analysis. When we applied these methods to the real-world DILI chronicity dataset, MARS and MDR identified SNP–SNP interactions that showed improved performance regarding the association with DILI chronicity as compared to their individual SNP counterparts. This finding demonstrated that MARS and MDR are superior to the traditional logistic regression approach for identifying SNP–SNP interactions in complex diseases such as chronic DILI.

We also developed a simple decision tree model by using the SNPs from the interactions identified by MARS and MDR, and most of these SNPs were mechanistically relevant, either related to the bilirubin uptake or to immunological responses. These SNPs will be further investigated and validated by including non-genetic factors such as drug properties and responses as co-factors. The identification of SNP–SNP interactions for disease susceptibility has become increasingly important, and our evaluated methods for determining potential genetic risk of the synergetic SNP interactions will potentially benefit the diagnostic test of chronic DILI.

## Figures and Tables

**Figure 1 ijerph-18-10603-f001:**
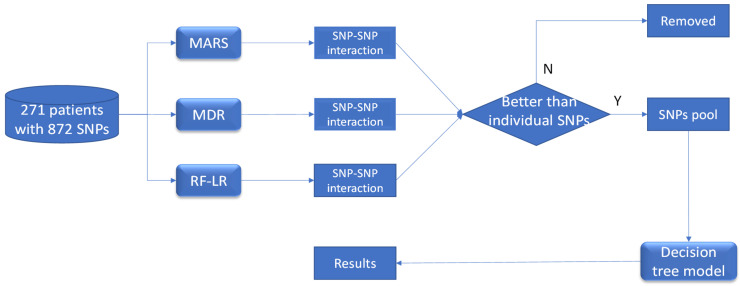
The diagram of the working flow to identify SNP–SNP interaction as a potential risk factor for chronic drug-induced liver injury. Specifically, multivariate adaptive regression spline (MARS), multifactor dimensionality reduction (MDR) and Random Forest plus logistic regression (RF-LR) were used to identify the SNP–SNP interaction terms linked to chronic DILI. Only the SNP–SNP interactions which had better association with DILI chronicity than individual SNPs were kept. All these SNPs of the selected interactions were pooled together as the candidate predictors, and a decision tree model is developed using classification and regression trees algorithm.

**Figure 2 ijerph-18-10603-f002:**
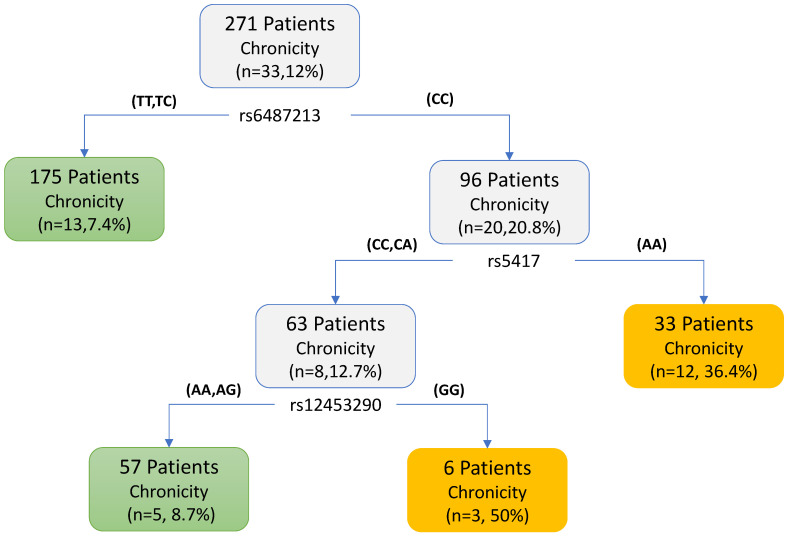
A decision tree model was developed to predict chronicity of drug-induced liver injury for 271 patients. Specifically, in the first layer, the genotype TT and TC of SNP rs6487213 will be assigned as acute, while the genotype CC will be continued to the next layer. In the second layer, the patient with the genotype AA of rs5417 will be assigned as chronic, otherwise, it will need further consideration. In the third layer, if the genotype GG of SNP rs12453290 was determined, the patient will be assigned as chronic and otherwise as acute.

**Table 1 ijerph-18-10603-t001:** Recovery rate of the known SNP4–SNP9 interaction by using three machine learning approaches in the original simulation dataset and permutation tests.

	Recovery Rate of the Known SNPs Interaction		
	MARS *	MDR *	RF-LR *
Original simulation dataset	100%	100%	100%
Permutation test 1 (2 of 250 observations permutated)	100% (100/100)	100% (100/100)	100% (100/100)
Permutation test 2 (10 of 250 observations permutated)	100% (100/100)	97% (97/100)	100% (100/100)

* MARS: Multivariate Adaptive Regression Splines; MDR: Multifactor Dimensionality Reduction; RF-LR: Random Forest plus Logistic Regression.

**Table 2 ijerph-18-10603-t002:** Individual SNPs and SNP–SNP interaction analysis for their association with chronic drug-induced liver injury.

Individual SNP Analysis				
Individual SNPs or SNP–SNP Interaction	Genotypes	Acute N (%)	Chronicity N (%)	Odds Ratio (95% Confidence Intervals, *p* Value)
rs6487213	CC	76 (79.2)	20 (20.8)	3.28 (1.57–7.09, *p* = 0.002)
	CT or TT	162 (92.6)	13 (7.4)	
rs5417	AA	74 (79.6)	19 (20.4)	3.01 (1.44–6.43, *p* = 0.004)
	CA or CC	164 (92.1)	14 (7.9)	
rs7658048	AG	113 (86.3)	18 (13.7)	1.33 (0.64–2.79, *p* = 0.448)
	AA or GG	125 (89.3)	15 (10.7)	
rs12453290	AA	105 (84.7)	19 (15.3)	1.72 (0.83–3.65, *p* = 0.149)
	GA or GG	133 (90.5)	14 (9.5)	
rs3785157	CC	106 (85.5)	18 (14.5)	1.49 (0.72–3.14, *p* = 0.282)
	TC or TT	132 (89.8)	15 (10.2)	
SNP–SNP interaction analysis				
MARS analysis				
rs6487213 + rs3785157	CC and CC	32 (69.6)	14 (30.4)	4.74 (2.14–10.39, *p* < 0.001)
	Others	206 (91.6)	19 (8.4)	
MDR analysis				
rs5417 + rs7658048 + rs12453290	AA and AG and AA	12 (66.7)	6 (33.3)	4.19 (1.36–11.74, *p* = 0.008)
	Others	226 (89.3)	27 (10.7)	
Random Forest plus logistic regression				
rs5417 + rs3785157	AA and CC	36 (78.3)	10 (21.7)	2.44 (1.03–5.45, *p* = 0.034)
	Others	202 (89.8)	23 (10.2)	

## Data Availability

Restrictions apply to the availability of these data. Data was obtained from International Serious Adverse Event Consortium and are available with the permission of International Serious Adverse Event Consortium.

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
