# Peer review of "Machine Learning to Identify Interaction of Single-Nucleotide Polymorphisms as a Risk Factor for Chronic Drug-Induced Liver Injury"

_ijerph, 2021, doi:10.3390/ijerph182010603_

Round 1
Reviewer 1 Report
Report on “Machine Learning to Identify Interactions of Single-Nucleotide Polymorphisms for Predicting Chronic Drug-Induced Liver Injury” by Minjun Chen et al.
The manuscript investigated three machine learning approaches: Multifactor Dimensionality Reduction (MDR), Random Forest plus Logistic Regression 68 (RF+LR), and Multivariate Adaptive Regression Splines (MARS), to evaluate their performance in identifying SNP interactions, and applied them to identifying the SNP interactions associated with DILI chronicity. I think that the methodology is quite clear and the manuscript is well written. This could be a good exercise to show how advanced statistical methodology could help to identify new predictors.
Concerns:
- Although authors declare in the introduction sections that “SNPs themselves come in huge numbers, sometimes in the hundreds of thousands; and including their interactions, the number of factors which need to be considered could expand exponentially”, they applied MLR algorithms in not so huge datasets. I think they have to include a comment on the limitations of the study.
- As this a comparison between algorithms in order to reduce the dimensionality of the problem, it is interesting to analyze the difference between them in computational times.
- The authors introduce a key point in multivariate modelling that is the interplay between variables, measuring as interactions. I agree with them, although probably it is necessary some comments to another important issue in multivariate modelling as is the correlation among markers (see for example Bansal A, Pepe MS. When does combining markers improve classification performance and what are implications for practice? Stat Med. 2013 May 20;32(11):1877-92. doi: 10.1002/sim.5736.)
Author Response
The manuscript investigated three machine learning approaches: Multifactor Dimensionality Reduction (MDR), Random Forest plus Logistic Regression 68 (RF+LR), and Multivariate Adaptive Regression Splines (MARS), to evaluate their performance in identifying SNP interactions, and applied them to identifying the SNP interactions associated with DILI chronicity. I think that the methodology is quite clear and the manuscript is well written. This could be a good exercise to show how advanced statistical methodology could help to identify new predictors.
We greatly appreciated the reviewer’s positive comments.
Concerns:
1. Although authors declare in the introduction sections that “SNPs themselves come in huge numbers, sometimes in the hundreds of thousands; and including their interactions, the number of factors which need to be considered could expand exponentially”, they applied MLR algorithms in not so huge datasets. I think they have to include a comment on the limitations of the study.
As suggested, we added this comments in discussion.
2. As this a comparison between algorithms in order to reduce the dimensionality of the problem, it is interesting to analyze the difference between them in computational times.
Thanks for the suggestion, we have discussed the difference of computational times of these approaches in the discussion.
3. The authors introduce a key point in multivariate modelling that is the interplay between variables, measuring as interactions. I agree with them, although probably it is necessary some comments to another important issue in multivariate modelling as is the correlation among markers (see for example Bansal A, Pepe MS. When does combining markers improve classification performance and what are implications for practice? Stat Med. 2013 May 20;32(11):1877-92. doi: 10.1002/sim.5736.)
We agreed with the reviewer’s comments. The comments about the correlations among markers and the suggested reference have been added in the introduction.
Reviewer 2 Report
This study aimed to identify appropriate statistical methods to investigate gene-gene and/or gene-environment interactions that impact Drug-induced liver injury (DILI) susceptibility. Three machine learning approaches including MDR, RF+LR, and MARS were used.
Drug-induced liver injury (DILI) is one of the main reasons for halting drug development processes. It is important to identify risk factors for preventing and reducing the liability of this disease. This study was designed and written well. But I have doubt about the SNPs selection.
Line 205-211, “As shown in Table 2, MARS and RF+LR identified two-way SNP interactions with rs6487213 x rs3785157 and rs5417 x rs3785157, while MDR identified a three-way interaction among rs7658048, rs5417 and rs12453290.” Before the description, the three SNPs (i.e., rs6487213, rs5417 and rs12453290) were not introduced in detail. How these SNPs were identified for the models?
In addition, the discussion section was simple and not enough for the significance and application of the machine learning approaches.
Author Response
This study aimed to identify appropriate statistical methods to investigate gene-gene and/or gene-environment interactions that impact Drug-induced liver injury (DILI) susceptibility. Three machine learning approaches including MDR, RF+LR, and MARS were used.
Drug-induced liver injury (DILI) is one of the main reasons for halting drug development processes. It is important to identify risk factors for preventing and reducing the liability of this disease. This study was designed and written well. But I have doubt about the SNPs selection.
We thank the reviewer’s positive comments
Line 205-211, “As shown in Table 2, MARS and RF+LR identified two-way SNP interactions with rs6487213 x rs3785157 and rs5417 x rs3785157, while MDR identified a three-way interaction among rs7658048, rs5417 and rs12453290.” Before the description, the three SNPs (i.e., rs6487213, rs5417 and rs12453290) were not introduced in detail. How these SNPs were identified for the models?
As suggested, we have revised the sentences in the results to address the reviewer’s comments
In addition, the discussion section was simple and not enough for the significance and application of the machine learning approaches.
We appreciated this important comment and have added a paragraph to compare the significance and limitations of these machine learning approaches in the discussion.
Reviewer 3 Report
In this paper the authors aimed to identify appropriate statistical methods to investigate gene-gene and/or gene-environment interactions that impact DILI susceptibility.
The authors used pre-packaged packages to perform analyzes on already available datasets. The right emphasis is not given to the data used for simulations. The first dataset is provided as an example by the software package used, while the other is extracted from a site that is not working. The technologies used (Machine Learned-based methods) have already been adopted to make similar analyzes. Furthermore, the authors present the work very superficially, without describing in detail the technologies used. Finally, the bibliography used is not adequate both in number and in quality.
Section 1 must be improved. You should introduce the problem in more detail so that the reader is immediately clear about the purpose of your study. Specify better the essential elements of the problem. You should add more information in the introductory part, you should add other works that have also addressed the problem. You must properly introduce your work, specify well what were the goals you set yourself and how you approached the problem. At the end of the section, add an outline of the rest of the paper, in this way the reader will be introduced to the content of the following sections.
Section 2 must be improved. Introduce better the machine learning techniques that you will use for your models (for example Random Forest ands Logistic Regression). In this section you present the basics of your work. You should spend more time introducing the basic topics. Since you have used pre-packaged packages for your analyzes, the data you have used is of great importance. It is not clear the difference between simulation study dataset and chronic DILI dataset. Why did you use them both? Explain it before introducing them, also you should explain in detail which features you used to train the models.
Section 3 must be improved. The evaluation metrics used are not adequately introduced. Justify the 100% of successfully identified the known SNP interactions. You haven't gotten into data splitting. Has cross validation been applied?
Section 5 must be improved. Paragraphs are missing where the possible practical applications of the results of this study are reported. What these results can serve the people, it is necessary to insert possible uses of this study that justify their publication. They also lack the possible future goals of this work. Do the authors plan to continue their research on this topic?
37) Check the format of reference.(1) . I have seen that you use this format for all references. The format is [1]
48) Do not use abbreviation such as e.g. I have seen that you often use this abbreviation, so I will not repeat this advice again, it also applies to the other occurrences.
51-52) Add references to support these statements.
73) Introduce your goals.
74-75) logistic regression. Introduce adequately the topic
76) Check the equation format
78) Check the equation format
79) Introduce adequately MARS , MDR ,and RF+LR
94) Check the link format
102) Do not use abbreviation such as i.e. I have seen that you often use this abbreviation, so I will not repeat this advice again, it also applies to the other occurrences.
104) Check the equation format
121)Introduce adequately the Generalized-Cross-Validation
123) Check the equation format
165) Check the link: don’t work
185-195) Justify the 100% of successfully identified the known SNP interactions. You haven't gotten into data splitting. Has cross validation been applied?
Author Response
In this paper the authors aimed to identify appropriate statistical methods to investigate gene-gene and/or gene-environment interactions that impact DILI susceptibility.
The authors used pre-packaged packages to perform analyzes on already available datasets. The right emphasis is not given to the data used for simulations. The first dataset is provided as an example by the software package used, while the other is extracted from a site that is not working. The technologies used (Machine Learned-based methods) have already been adopted to make similar analyzes. Furthermore, the authors present the work very superficially, without describing in detail the technologies used. Finally, the bibliography used is not adequate both in number and in quality.
We greatly appreciated the reviewer’s comments and critics, which are very helpful to improve our manuscript. Besides addressing the reviewer’s comments as below, we have added more of quality references in the revision.
Section 1 must be improved. You should introduce the problem in more detail so that the reader is immediately clear about the purpose of your study. Specify better the essential elements of the problem. You should add more information in the introductory part, you should add other works that have also addressed the problem. You must properly introduce your work, specify well what were the goals you set yourself and how you approached the problem. At the end of the section, add an outline of the rest of the paper, in this way the reader will be introduced to the content of the following sections.
As suggested, we reorganized the introduction to better clarify our goal and the question we encountered, and how we would like to address this problem. We also summarized and cited other works for identifying SNP-SNP interactions to assess disease susceptibility. At the end of introduction, we provided a brief outline of the rest of this paper to help readers better follow our manuscript.
Section 2 must be improved. Introduce better the machine learning techniques that you will use for your models (for example, Random Forest and Logistic Regression). In this section you present the basics of your work. You should spend more time introducing the basic topics. Since you have used pre-packaged packages for your analyzes, the data you have used is of great importance. It is not clear the difference between simulation study dataset and chronic DILI dataset. Why did you use them both? Explain it before introducing them, also you should explain in detail which features you used to train the models.
In this section, we first reordered the machine learning approaches by setting MARS and MDR ahead of logistical regression because the first two approaches perform better. We then added an introduction about the use of the simulated dataset and the chronic DILI dataset. We also clarified about the use of SNPs for model training.
Section 3 must be improved. The evaluation metrics used are not adequately introduced. Justify the 100% of successfully identified the known SNP interactions. You haven't gotten into data splitting. Has cross validation been applied?
Thanks for the comments. We added the explanation about the evaluation metrics in the simulated dataset. In the DILI chronicity data, we used hold-out validation instead of cross-validation. We have added an external validation set with N=76 patients to assess the model’s predictive performance.
Section 5 must be improved. Paragraphs are missing where the possible practical applications of the results of this study are reported. What these results can serve the people, it is necessary to insert possible uses of this study that justify their publication. They also lack the possible future goals of this work. Do the authors plan to continue their research on this topic?
We have revised the section 5 as suggested. The potential practical application of our findings and the future work to continue this research were added.
37) Check the format of reference.(1) . I have seen that you use this format for all references. The format is [1]
All of the references were reformatted as the style of numbered such as [1]
48) Do not use abbreviation such as e.g. I have seen that you often use this abbreviation, so I will not repeat this advice again, it also applies to the other occurrences.
The abbreviation such as e.g. and i.e. are removed from the revision.
51-52) Add references to support these statements.
Added as suggested.
73) Introduce your goals.
Added
74-75) logistic regression. Introduce adequately the topic
Added
76) Check the equation format
Checked
78) Check the equation format
Checked
79) Introduce adequately MARS , MDR ,and RF+LR
We added some more introduction and references for these three approaches in the revision.
94) Check the link format
The link was removed now.
102) Do not use abbreviation such as i.e. I have seen that you often use this abbreviation, so I will not repeat this advice again, it also applies to the other occurrences.
We removed these abbreviations in the revision.
104) Check the equation format
Checked.
121)Introduce adequately the Generalized-Cross-Validation
We revised this paragraph to better describe the generalized cross-validation.
123) Check the equation format
Checked
165) Check the link: don’t work
ISAEC stopped the web service recently, and we have removed the link.
185-195) Justify the 100% of successfully identified the known SNP interactions. You haven't gotten into data splitting. Has cross validation been applied?
Please check our response above.
Round 2
Reviewer 2 Report
The authors have made a lot of modifications according to the reviews' comments, and the manuscript has been greatly improved over the original version.
But some paragraphs still need to be revised. For example the last paragraph in the introduction section“In this study, ......... and all of them performed well. Next, these three methods identified the association of SNP interactions with DILI chronicity. The SNP-SNP interactions revealed by MARS and MDR, but not logistical regression, showed .......... these SNP interactions may help the diagnosis for DILI chronicity.”
As important results and conclusions of the manuscript, they should not appear in the introduction section, but the Results and Discussion section.
Author Response
The authors have made a lot of modifications according to the reviews' comments, and the manuscript has been greatly improved over the original version.
But some paragraphs still need to be revised. For example the last paragraph in the introduction section“In this study, ......... and all of them performed well. Next, these three methods identified the association of SNP interactions with DILI chronicity. The SNP-SNP interactions revealed by MARS and MDR, but not logistical regression, showed .......... these SNP interactions may help the diagnosis for DILI chronicity.”
As important results and conclusions of the manuscript, they should not appear in the introduction section, but the Results and Discussion section.
We greatly appreciated the reviewer’s comments and have rewritten the last paragraph of the introductions as below. We also carefully review other parts of the manuscript.
In this study we aim to investigate SNP-SNP interaction as a potential risk factor for predicting DILI chronicity using machine learning approaches. We will briefly introduce three machine learning methods used, including MARS, MDR, and logistic regression. Then, we will use a simulated data with a known pairwise SNP-SNP interaction to evaluate the accuracies and robustness of these methods. Next, we will apply the three methods to identify SNP interactions associated with chronic DILI. Finally, we will evaluate the predictive performance of the identified SNPs by using a simple decision tree model to assess whether increased risk of DILI chronicity was associated with the presence of these SNP interactions.
Reviewer 3 Report
The authors only partially addressed the suggestions provided by the reviewer. In some cases they claimed to have addressed it but the paper has not been modified. For example for the format of the equations which is not in line with that required by the journal. The authors said they did a check but nothing has changed. Unfortunately, I noticed several formatting problems also in this second version of the paper, it seems that the authors are in a hurry to publish the work. A thorough check of the paper format (text, formulas and Figures) is necessary.
The greatest criticality of the work is to demonstrate the originality of the study. The authors used a dataset to accompany a software that already carries out these analyzes by itself, indeed it uses this data as an example. Then the authors must clearly describe the novelty their study brings.
Furthermore, the description of the dataset is lacking. Authors must clearly indicate which variables were used as inputs and indicate which scientific evidence led to their use. Add a diagram to show this information.
The authors must specify in detail the software platform used for the implementation of the algorithms. if pre-packaged packages have been used, they must describe the characteristics of the libraries.
Finally, the proposed bibliography is insufficient, both in number and in quality of the works.
Try to enrich the captions of the figure, the reader should be able to read the figure without the need to retrieve the information in the paper. Try to summarize the essential parts of the Figure and what you want to explain with it.
Minor revision
Check the format of the title text (Interactions)
At the end of the Introduction section, add an outline of the rest of the paper, in this way the reader will be introduced to the content of the following sections.
Check all the equations format. This is not the journal format.
135-141) Check the text format
Author Response
The authors only partially addressed the suggestions provided by the reviewer. In some cases they claimed to have addressed it but the paper has not been modified. For example for the format of the equations which is not in line with that required by the journal. The authors said they did a check but nothing has changed. Unfortunately, I noticed several formatting problems also in this second version of the paper, it seems that the authors are in a hurry to publish the work. A thorough check of the paper format (text, formulas and Figures) is necessary.
We apologize to the reviewer for our misunderstanding of some comments. In the previous revision, we don’t realize the equations were not written in the format required by the journal but focus on checking the misspelling within the equations. Here we have re-checked the guidance of the journal and communicated with the editor about the requirement of format of the equations. In this revision, we have rewritten all the equations and related symbols in the text using the equation module in the Microsoft Office Word, and the format is revised following the template provided by the editor.
We are sorry to leave some formatting issues unsolved in the previous revision. Although it should not be an excuse, we wish to have additional time beyond the 10 days the editor gave to conduct a more thorough check of the language and formats. In this revision, we have tried our best to check the manuscript formats as much as possible.
The greatest criticality of the work is to demonstrate the originality of the study. The authors used a dataset to accompany a software that already carries out these analyzes by itself, indeed it uses this data as an example. Then the authors must clearly describe the novelty their study brings.
We greatly appreciated the reviewer’s comments. In this revision, we have modified the title as “Machine Learning to Identify Interaction of Single-Nucleotide Polymorphism as a Risk Factor for Chronic Drug-Induced Liver Injury” to better reflect the novelty of this study. We also reorganize the introduction and add the references about the genetic study of drug-induced liver injury. Besides, we add the statement that “However, most of these studies investigated the association of each SNP individually [20, 21]; SNP-SNP interaction as a potential risk factors for drug-induced liver injury was seldom reported.” In the beginning of the last paragraph in the introduction, we reinstated the novelty of our work as “In this study we aim to investigate SNP-SNP interaction as a potential risk factor for predicting DILI chronicity using machine learning approaches.” We hope these newly added statements would better inform the readers about the novelty of this study.
Furthermore, the description of the dataset is lacking. Authors must clearly indicate which variables were used as inputs and indicate which scientific evidence led to their use. Add a diagram to show this information.
As suggested, we reorganized the method & material part and put the DILI chronicity cohort as the first part of this section. We also added more detail about how the dataset is generated and cited the reference about the methodologies as detailed below
DNA preparation is described here [29] and standard quality control procedures were followed [30] Samples were genotyped using the HumanOmniExpressExome-8v1 and the Illumina HumanCoreExome-12 v1.0 BeadChip, from which 872 SNPs associated with the genes in bile acid pathways were retrieved for analysis. Bile acid pathways were downloaded from the Molecular Signature Database (MSigDB) [31] and included pathways involved in bile acid synthesis, recycling, and transport.
Additionally, we added a new section of “data analysis” to describe our modeling process. We added a paragraph and a diagram to explain how we process the data and build the model. Now it reads as
We also applied these machine learning approaches to a real-life dataset of DILI chronicity. Figure 1 briefly illustrated the pipeline for detecting the potential risk factors of SNP-SNP interactions which are associated with chronic drug-induced liver injury. Firstly, based on the dataset of 271 patients and 872 SNPs three machine learning methods including MARS, MDR and Random Forest plus logistical regression were utilized to identify the mostly relevant SNP-SNP interactions linked to DILI chronicity. Next, only the SNP-SNP interactions have the better association with DILI chronicity, which is measured by odd ratio, than the individual SNPs were considered. Finally, The SNPs from the selected interactions were pooled together as candidate predictors, and then a decision tree model using classification and regression trees (CART) algorithm is developed.
The authors must specify in detail the software platform used for the implementation of the algorithms. if pre-packaged packages have been used, they must describe the characteristics of the libraries.
As suggested, we added the more detail of the software platform and algorithms used in the revision. The references for these methods were also added. Now it reads as
All analyses except those specially mentioned were performed using R (version 3.6.1) [39] and the MDR package [40] for multifactor dimensionality reduction approach, the randomForest package [41] for Random Forest algorithm, the Rpart package [42] for decision tree model, and the Stats package for logistic regression and Fisher exact test. Multivariate adaptive regression spline approach was implemented using the MARS engine of Salford Predictive Modeler 8.0 from MinTab [43].
Finally, the proposed bibliography is insufficient, both in number and in quality of the works.
As suggested, we have added new references to reflect the progresses of drug-induced liver injury and genetic studies, most of which were published in last 5 years and came from high-quality journals such as Nature Review series, New England Journal of Medicine, Gastroenterology, Hepatology, Journal of Hepatology. We also added the references for the experimental methods and machine learning models we used. Now the revision has cited a total of 55 references.
Try to enrich the captions of the figure, the reader should be able to read the figure without the need to retrieve the information in the paper. Try to summarize the essential parts of the Figure and what you want to explain with it.
Thank the reviewer’s suggestion. Now we added more of essential information to the captions of figure 1 and 2 to allow readers to better read them.
Figure 1. The diagram of the working flow to identify SNP-SNP interactions as potential risk factors for chronic drug-induced liver injury. Specifically, multivariate adaptive regression spline (MARS), multifactor-dimensionality reduction (MDR) and Random Forest plus logistical regression (RF-LR) were used to identify the SNP-SNP interactions linked to DILI chronicity. Only the SNP-SNP interactions which had better association with DILI chronicity than individual SNPs were kept. All these SNPs of the selected interactions were pooled together as the candidate predictors, and a decision tree model is developed using classification and regression trees (CART) algorithm.
Figure 2. A decision tree model was developed to predict chronicity of drug-induced liver injury for 271 patients. Specifically, in the first layer, the genotype TT and TC of SNP rs6487213 will be assigned as acute, while the genotype CC will be continued to the next layer. In the second layer, the patient with the genotype AA of rs5417 will be assigned as chronic, otherwise, it will need further consideration. In the third layer, if the genotype GG of SNP rs12453290 was determined, the patient will be assigned as chronic and otherwise as acute.
Minor revision
Check the format of the title text (Interactions)
We have rewritten the title. Now it reads as
Machine Learning to Identify Interaction of Single-Nucleotide Polymorphisms as a Risk Factor for Chronic Drug-Induced Liver Injury
At the end of the Introduction section, add an outline of the rest of the paper, in this way the reader will be introduced to the content of the following sections.
As suggested, we have added an outline at the end of the introduction. Now it reads as
In this study we aim to identify SNP-SNP interaction as a potential risk factors for predicting DILI chronicity using machine learning approaches. We will briefly introduce three machine learning methods used, including MARS, MDR, and logistic regression. Then, we will use a simulated data with known pairwise SNPs to evaluate the accuracy and robustness of these methods. Next, we will apply the three methods to identify the SNP interactions associated with to chronic DILI. Finally, we will evaluate the predictive performance of these identified SNPs by using a simple decision tree model to assess the risk of DILI chronicity associated with the presence of these SNP interactions.
Check all the equations format. This is not the journal format.
We have rewritten the 4 equations and other symbols in the text using the equations module in Words and changed the format following the template provided by the editor. Sorry for the misunderstanding.
135-141) Check the text format
We rewrote the part of line 135-141. Now it reads as
Its main concept is to utilize a specially designed strategy to reduce multi-locus information to a one-dimensional model. In this strategy, a subset of n genetic factor is firstly selected, and then these n-genetic factors and their possible multifactor classes are represented in n-dimensional space, in which each multifactor cell will be labeled as high-risk or low-risk group. In this way, the case-control model for multi-locus genotypes will be converted into classification of high-risk versus and low-risk, which reduces the n-dimensional model to a one-dimensional model. In next step, a cross-validation and its estimated prediction error will be used to evaluate the selected n-genetic factor. For each n genetic factor combination, a single model that minimizes the average classification error in the cross-validation training sets is selected. This will result in a list of best models, one for each value of n genetic factor. Among these best classification models, the combination of the genetic factors and the model they built that minimizes the average prediction error across the prediction errors in the cross-validation testing sets will be selected as the final one.